# Immunoassay Urine Drug Testing among Patients Receiving Opioids at a Safety-Net Palliative Medicine Clinic

**DOI:** 10.3390/cancers15235663

**Published:** 2023-11-30

**Authors:** John M. Halphen, Joseph A. Arthur, Soraira Pacheco, Linh M. T. Nguyen, Nikitha N. Samy, Nathaniel R. Wilson, Gregory Sattler, Shane E. Wing, Rex A. D. Paulino, Christine Feng, Pulin Shah, Simbiat Olayiwola, Bradley Cannell, Supriyanka Addimulam, Riddhi Patel, David Hui

**Affiliations:** 1Joan and Stanford Alexander Division of Geriatric and Palliative Medicine, McGovern Medical School, UTHealth, Houston, TX 77030, USA; soraira.pacheco@uth.tmc.edu (S.P.); linh.m.nguyen@uth.tmc.edu (L.M.T.N.); simbiat.k.olayiwola@uth.tmc.edu (S.O.); 2Department of Palliative Care, Rehabilitation and Integrative Medicine, MD Anderson Cancer Center, Houston, TX 77030, USA; jaarthur@mdanderson.org (J.A.A.); dhui@mdanderson.org (D.H.); 3McGovern Medical School, The University of Texas Health Science Center at Houston, Houston, TX 77030, USAnathaniel.r.wilson@uth.tmc.edu (N.R.W.); gregory.sattler@wacofamilymedicine.org (G.S.); shane.e.wing@uth.tmc.edu (S.E.W.); rexalvin.d.paulino@uth.tmc.edu (R.A.D.P.);; 4Department of Epidemiology, Human Genetics & Environmental Sciences, The University of Texas Health Science Center at Houston, Houston, TX 77030, USA; michael.b.cannell@uth.tmc.edu (B.C.); supriyanka.addimulam@gmail.com (S.A.); riddhi.r.patel@uth.tmc.edu (R.P.)

**Keywords:** immunoassay, urine drug test, opioid, nonmedical, cancer pain, education, safety-net hospital, palliative medicine

## Abstract

**Simple Summary:**

A urine drug test (UDT) is often used in the treatment of cancer pain to monitor compliance with opioid treatment. Two types of UDT are commonly for this purpose: the immunoassay test, and the mass spectrometry method. Only a few studies have examined the use of immunoassay UDT for cancer patients in palliative care clinics. In this study, we examined the frequency of immunoassay UDT abnormalities, and the factors associated with aberrant findings at a safety-net hospital palliative medicine clinic. Electronic medical records of 913 patients were reviewed. We found that 27% had aberrant UDT results; 35% of these were positive for cocaine. Non-Hispanic White race, history of illicit drug use, and history of marijuana use were associated with an aberrant finding. Despite limitations of immunoassay UDT, it could detect aberrant drug-taking behaviors in a significant number of patients. These findings support the utility of immunoassay UDT in clinical settings with less resources.

**Abstract:**

Background: Few studies have examined the use of immunoassay urine drug testing of cancer patients in palliative care clinics. Objectives: We examined the frequency of immunoassay urine drug test (UDT) abnormalities and the factors associated with aberrancy at a safety-net hospital palliative medicine clinic. Methods: A retrospective review of the electronic medical records of consecutive eligible patients seen at the outpatient palliative medicine clinic in a resource-limited safety-net hospital system was conducted between 1 September 2015 and 31 December 2020. We collected longitudinal data on patient demographics, UDT findings, and potential predictors of aberrant results. Results: Of the 913 patients in the study, 500 (55%) underwent UDT testing, with 455 (50%) having the testing within the first three visits. Among those tested within the first three visits, 125 (27%) had aberrant UDT results; 44 (35%) of these 125 patients were positive for cocaine. In a multivariable regression model analysis of predictors for aberrant UDT within the first three visits, non-Hispanic White race (odds ratio (OR) = 2.13; 95% confidence interval (CI): 1.03–4.38; *p* = 0.04), history of illicit drug use (OR = 3.57; CI: 1.78–7.13; *p* < 0.001), and history of marijuana use (OR = 7.05; CI: 3.85–12.91; *p* < 0.001) were independent predictors of an aberrant UDT finding. Conclusion: Despite limitations of immunoassay UDT, it was able to detect aberrant drug-taking behaviors in a significant number of patients seen at a safety-net hospital palliative care clinic, including cocaine use. These findings support universal UDT monitoring and utility of immunoassay-based UDT in resource-limited settings.

## 1. Introduction

Patients treated with opioids for cancer pain in palliative care outpatient clinics may be at a high risk for nonmedical opioid use (NMOU) [1] and substance use disorder. NMOU [2] refers to misuse of opioids to self-treat non-pain symptoms, concurrent use of illicit drugs, diversion to unintended users, and varying degrees of opioid use disorder. It is also characterized by behaviors such as excessive, unjustifiable use of opioids or self-escalation of opioid dosage. NMOU is associated with a number of negative outcomes for patients and others in the community, including increased morbidity, opioid-related overdose death, and involvement in illegal activities [3]. Substance use disorders have been associated with social instability [4,5] and poor symptom control [6], and may potentially contribute to poor patient adherence to cancer treatments. Cancer patients with substance use disorders have two potentially fatal and disabling conditions, both of which require the attention of clinicians [7,8,9]. It is necessary to effectively screen for substance use disorder and monitor opioid use in palliative care clinics in order to ensure early identification and management of such complications. This has become particularly more relevant in palliative oncology settings because with the early integration of the palliative care model into oncologic care [10,11], palliative care clinicians are caring for patients earlier in the disease trajectory and therefore encountering an increasing number of patients with chronic pain receiving opioids. Clinical evidence suggests that one in five of such patients might be at risk for NMOU [1,2,12,13,14].

Prescribers of opioids have long been required by federal and state law to use caution and ensure that the medications are prescribed and used appropriately. This includes careful screening and monitoring of patients at risk for NMOU, as well as timely identification of those who are actively engaging in NMOU behaviors and substance misuse [15,16]. Urine drug testing for drugs of abuse, and for the presence or absence of the prescribed opioids, is a risk assessment measure often employed in the treatment of chronic cancer and non-cancer pain [12,17]. There are two types of a urine drug test (UDT) commonly employed for this purpose, the immunoassay test, and the more expensive, specific, and sensitive mass spectrometry methods, which may be used for initial screening or to confirm a positive result on the UDT. Due to the expense of mass spectrometry testing, it may not be affordable in resource-limited clinics. It takes a longer time to receive the results of the mass spectrometry testing and clinical treatment decisions may have to rely on the UDT in some circumstances.

A majority of studies on UDT have reported on the use of mass spectrometry; few studies have examined how the use of immunoassay UDT could inform clinical practice. This paper reports on the results of a study of immunoassay UDT conducted in an ambulatory palliative medicine clinic located in a resource-limited safety-net county hospital caring for predominantly indigent and uninsured patients. The objective of this study was to examine the frequency of UDT abnormalities found with the immunoassay test. We also examined the patient characteristics and factors associated with the UDT results that were considered aberrant findings by the clinical team. Successful examination of UDT abnormality rates using the immunoassay test will underscore its significance and utility in routine opioid therapy especially in resource-limited settings where this test might be the only available or most viable option.

## 2. Materials and Methods

### 2.1. Study Participants and Procedure

We conducted a retrospective review of the electronic medical records of consecutive eligible patients seen at the outpatient palliative medicine clinic at Lyndon B. Johnson General Hospital (LBJ) in Houston, Texas, between 1 September 2015 and 31 December 2020. LBJ is a safety-net hospital that serves predominantly low-income and uninsured patients. Approximately 85% of them are either uninsured or underinsured. A significant proportion of its revenue is generated from Medicaid Supplemental Programs. In Fiscal Year 2021, it provided over USD 720 million in charity care [18]. Patients were eligible for the study if they were 18 years of age or older, had a diagnosis of cancer (with or without active disease), and were receiving opioids for cancer-related pain at any time during the study period. The study was approved by the institutional review board of UTHealth and Harris Health Systems.

### 2.2. Data Collection

Patients’ baseline demographic and clinical characteristics were obtained within the first three clinic visits. These included patient age; sex; race and ethnicity; marital status; cancer type; and cancer stage. Also obtained within the first three clinic visits were pertinent risk factors for nonmedical opioid use such as history of illicit drug use; history of tobacco use; history of alcohol use; history of depression; history of bipolar disorder; history of schizophrenia; family history of illicit drug use; personal history of criminal activity (other than marijuana use); and contact with persons involved in criminal activity (other than marijuana use). Information regarding their opioid intake at the time of urine testing was obtained to assist with determining the morphine equivalent daily dose (MEDD) and facilitating interpretation of the UDT results. Weekly meetings were held among the clinician investigators to ensure uniformity in the data collection process. Efforts were made during the data collection process to maintain the confidentiality and privacy of study subjects in view of the sensitive nature of the health information obtained.

### 2.3. Clinic Process and Instruments

As part of the standard procedure in the clinic, patients receiving chronic opioid therapy were screened using risk assessment questions and the state prescription drug monitoring program (PDMP) database. Prior to opioid initiation, clinicians were encouraged to ask patients to complete a written pain treatment agreement and provide a verbal consent. Patients perceived to be at a high risk for NMOU based on the risk assessment tools and clinical interviews were monitored more closely on an ongoing basis including close observation of certain behavioral patterns suggestive of NMOU. Clinicians were encouraged and reminded to routinely obtain a baseline UDT within the first three clinic visits on every patient receiving opioids. If the clinician believed that the patient was at an elevated risk of NMOU, risk mitigation measures were implemented such as increasing the frequency of visits to the clinic; more cautious and limited opioid prescription; more frequent urine drug testing; intensive counselling; and referral to psychology and psychiatry as available.

### 2.4. The Urine Drug Test

The specific UDT reagents used in this study were manufactured by either Siemens Vista or Beckman Coulter. These were immunoassay tests designed to screen for the presence of opiates, amphetamine, cocaine, phencyclidine, benzodiazepines, barbiturates, and cannabinoids. The UDT is based on the reaction of the drug being tested for (analyte) with a reagent that binds to it. The binding reagents may react with other substances in the urine besides the analyte, causing there to be false positive test results. This may happen when the reagent used to detect the presence of benzodiazepines registers a positive result when the patient consumes, for instance, sertraline instead of benzodiazepines. The other substances in the urine that cause false positive results on the UDT may vary by the manufacturer of the reagent used for the test [19]. The reagent used to detect opiates will bind to morphine or codeine but may fail to bind to synthetic or semi-synthetic opioids, because of their differences in chemical structure, thereby leading to false negative results. Because of the potential for false positive and negative results, confirmatory testing with mass spectrometry is often used along with the immunoassay UDT. Confirmatory testing was not available in this clinic during the study period, rendering the urine testing results presumptive and not conclusive, except for positive cocaine results, which were considered conclusive [20]. Because of the limitations of the type of UDT used, if a result indicated the consumption of an unauthorized or illicit substance, or a failure to detect a drug expected to be present, a review of the record and a conversation with the patient were used by the clinician to determine whether or not the UDT result reflected NMOU behavior. False positive and negative UDT result information was ultimately discarded and not utilized in guiding further therapeutic decisions for patients in the clinic. For this study, aberrant results were determined based on any of the following: unexpected presence of unprescribed opioids, unexpected absence of prescribed opioids, or presence of illicit drugs in the urine.

The prescribing clinicians consulted the literature, which described potential false positive or false negative results that may be encountered using the UDT [19,21,22]. For example, consumption of substances such as methylphenidate, trazodone, tyramine, labetalol, propranolol, bupropion, ephedrine, and pseudoephedrine may all lead to a positive reading for an amphetamine [23]. If the UDT was positive for benzodiazepines, consumption of sertraline could lead to a false positive. If the UDT was positive for cannabinoids, the consumption of dronabinol, non-steroidal anti-inflammatory drugs, or proton pump inhibitors might be the reason. A positive result on the UDT for phencyclidine might result from the consumption of dextromethorphan, diphenhydramine, or tramadol. If the UDT was positive for a barbiturate, the use of primidone, ibuprofen, or naproxen could be the reason [21]. However, the literature indicates that a positive result on the UDT for cocaine is reliable due to the consumption of cocaine, crack, coca leaf tea, or other cocaine-containing products. Thus, when the UDT was positive for cocaine, the urine test was always deemed aberrant. On the other hand, false negative results usually occur if a sample has a low drug concentration, or the test has a relatively high cut-off calibration [17]. Most immunoassays can only recognize classes of drugs (class assays) and are unable to distinguish between drugs in the same class. They also miss compounds such as oxycodone and synthetic opioids such as fentanyl and methadone [24]. All these can lead to false negative results. Details of substances that can potentially result in false positive and false negative results during clinical urine drug testing can be found elsewhere [23].

In order to minimize the impact on our interpretation of UDT results by potential patient dilution or substitution of samples submitted, a urinalysis, or a urine creatinine, was often ordered along with the UDT. A urine pH of 3–11, a specific gravity of 1.002–1.020, or creatinine of ≥5 mg/dL indicate an unadulterated urine sample. Collection of the samples was not observed, so use of another person’s urine was also possible.

### 2.5. Statistical Analysis

Descriptive statistics such as frequency and percentage for categorical data, and median with inter-quartile range (IQR) for continuous variables, were used to summarize the results. The chi-squared test or Fisher’s exact test were used to assess the association between categorical variables and UDT findings. The *t*-test was used to assess association between continuous variables and UDT findings. Univariate and multivariable logistic regression analyses were used to explore the demographics and clinical factors associated with aberrant UDT findings. Aberrant UDT (yes, no) was the main outcome. The independent variables evaluated were age, sex (male, female), race and ethnicity (non-Hispanic White, non-Hispanic Black, non-Hispanic and other race, and Hispanic and any race), marital status (married, single), cancer type (gastrointestinal, respiratory, gynecological, genitourinary, breast, head and neck, heme, and other), cancer stage (locally advanced, localized, recurrent, advanced, first line, and metastatic), history of illicit drug use (yes, no), history of marijuana use (yes, no), history of tobacco use (yes, no), history of alcohol use (yes, no), history of depression (yes, no), history of bipolar disorder (yes, no), history of schizophrenia (yes, no), family history of illicit drug use (yes, no), personal history of criminal activity (yes, no), and contact with persons involved in criminal activity (yes, no). A *p*-value cut-off <0.05 was considered statistically significant. The data were analyzed with STATA software, version 17 (Stata Corporation, College Station, TX, USA).

## 3. Results

Table 1 provides information on demographic and clinical characteristics of consecutive study patients seen at the palliative care clinic and those who underwent a baseline UDT within the first three clinic visits. Of 913 study patients seen in the clinic, 455 (50%) patients underwent a UDT within the first three visits and of those, 125 (27%) patients were found to have aberrant UDT results. The median age of patients seen in the clinic was 55 years. The majority were female (480, 53%), Hispanic, any race (425, 47%), and single (610, 67%). Half of the patients in the study did not receive a UDT within the first three visits.

Table 2 shows the frequency and percentage of patients who were seen in the clinic, underwent the UDT, and had aberrant UDT results during the clinic visits. The majority of patients seen in the clinic underwent at least one UDT (455 (50%) during the first three visits and 500 (55%) during the entire study period). The UDT was most frequently administered during the initial visit. Of the patients who had a UDT, 91% had the test within the first three visits. Approximately 27% and 29% of the tests were deemed aberrant within the first three clinic visits and during the entire study period, respectively. Aberrant results triggered a record review and conversation with the patient. None of the aberrant UDT results were caused by cross-reaction of prescribed or over-the-counter medications.

Figure 1 A depiction of the types and distribution of illicit substances present in the UDT of the patients tested during the study period. Of the 125 patients who had aberrant urine samples in the first three clinic visits, the following numbers of patients had positive results for common illicit drugs screened for: amphetamine (9, 7%); barbiturate (2, 2%); benzodiazepines (15, 12%); cannabinoids (87, 70%); cocaine (44, 35%); and PCP (3, 2%). The patients who had cocaine in the urine constituted 9.7% of all patients that had urine tested during the first three visits.

In a multivariable analysis of factors associated with the ordering of UDT (Table 3), the odds of ordering a UDT within the first three visits to the clinic decreased by 3% with each 1-year increase in the age (OR: 0.97; 95% CI: 0.96, 0.99). The odds of ordering a UDT among non-Hispanic Whites was 2.02 times (95% CI: 1.37, 2.98) and among non-Hispanic Blacks it was 1.86 times (95% CI: 1.30, 2.65) that of Hispanics. Moreover, patients with head and neck cancer had 2.18 (95% CI: 1.25, 3.79) times the odds of ordering a test than those with a gastrointestinal cancer. Patients with a locally advanced cancer stage had 58% (OR: 1.58; 95% CI: 1.10, 2.25) higher odds of undergoing a test as compared to those with metastatic cancer. Additionally, patients with a prior history of illicit drug use had 1.81 times (95% CI: 1.08, 3.04) and those with a history of marijuana use had 1.65 times (95% CI: 1.09, 2.50) the odds of undergoing UDT within the first three visits. Also, non-Hispanic Whites had about twice (OR: 2.13; 95% CI: 1.03, 4.38) the odds of aberrant results as compared to Hispanics.

## 4. Discussion

Half of the 913 patients with cancer pain included in this study underwent a UDT during the first three clinic visits; of these, 125 (27%) had aberrant UDT results. In total, 44 (35%) of these 125 patients had UDT results positive for cocaine. The number of patients found to have aberrant UDT results was high. Studies have found similar abnormal urine testing results in other palliative medicine clinics [2,25]. Overall, our study shows that patients with cancer and on opioids have a significant risk for NMOU that could be detected with immunoassay UDT in routine clinical practice [25,26]. Cancer patients, like the general population, may have pre-existing drug-related issues. This, coupled with the increased exposure to opioids for cancer pain management, increases the risk for NMOU [3,27]. It is important to note that the use of UDS in the palliative care setting might be of more relevance among ambulatory palliative care patients with relatively longer survival than among those close to the end of life who have a very short life expectancy.

The rate of cocaine use in our population was higher than some other studies involving populations with different socioeconomic factors [13]. In one study conducted at another palliative care clinic whose patients have a different demographic mix, with a high percentage of insured and racial-majority patients, 8.2% of these patients who underwent risk-based urine drug testing had cocaine in the urine [12] and only 1% of patients who were randomly selected for testing irrespective of risk tested positive for cocaine [28]. Our study was conducted in a safety-net palliative medicine clinic with predominantly ethnic and racial minorities where most of the patients were uninsured, underinsured, and had less resources. Of the 913 consecutive patients in our study, 47% were Hispanic and any race, 28% were Black non-Hispanic, and 3% were of other races and non-Hispanic. The percentage of patients testing positive for cocaine in the urine was higher than the risk-based testing protocol in the study mentioned above and would most likely have been even higher in our clinic if the testing was only directed toward patients with a high perceived risk of NMOU. Future studies are needed to ascertain whether less favorable social determinants of heath are key predictors of NMOU and OUD. Particularly, race or ethnicity has not been found to be a risk factor for opioid misuse although data in both palliative care and non-palliative care populations have revealed disparities in UDS ordering disproportionately affecting minoritized patients [29,30,31].

The high number of patients testing positive for cocaine is significant because concurrent use of cocaine or other illicit drugs and opioids can result in increased morbidity and mortality. Since substance use of one drug is often accompanied by misuse of other substances [32], cocaine use disorder might be indicative of problematic use of other substances as well as opioids. Cocaine is a highly addictive substance that can cause multiple serious health risks such as intravenous-drug-use-related infections, cognitive deficits, overdose deaths, as well as long-term cardiovascular, respiratory, gastrointestinal, and neurovascular complications [33,34]. More than 500,000 people sought medical attention in an emergency room (ER) for cocaine-related complications in 2011, accounting for over 40% of all ER visits involving illicit drug use [34]. These issues, coupled with the potential dangers of aberrant opioid use and complications from cancer and its treatment, pose significant problems for cancer patients with comorbid cocaine use disorder and NMOU. Early identification of patients who actively engage in cocaine use presents an opportunity for clinicians to take the necessary steps to avert potential harm to their patients and make the appropriate referrals of these patients to receive specialist care. This underscores the important role that immunoassay UDT might have in less-resourced populations where more expensive UDTs might not be available. It has been found that patients with substance use disorders who are undergoing cancer therapies and cancer symptom management face more challenges and may have worse outcomes [7,8,35]. Further studies are needed to investigate the impact of cocaine and other substance use disorders on the adherence and outcomes among patients undergoing anticancer therapies.

The immunoassay test, although limited, was useful in identifying a significant number of patients consuming illegal and unauthorized substances who required greater vigilance and assistance. The UDT, along with a review of the patient record and a conversation with the patient, can be of value although the results of the UDT are said to be presumptive except when cocaine is detected. The findings support the notion that this type of UDT may be useful as a routine risk mitigation tool in patients with chronic cancer pain. Entities such as the Centers for Disease Control explicitly excluded cancer-related chronic pain from their guidelines [36]. However, it is becoming more evident from multiple studies that urine drug testing is useful in chronic cancer-related pain. Also, universal screening of all patients for substance use disorder and NMOU with UDT in a palliative care clinic [35] would possibly reduce the potential negative impact of selective UDT testing on the physician–patient relationship. The patient is likely to see it as part of the clinic’s routine policy and not feel targeted if the UDT was required of all patients. The immunoassay UDT is inexpensive enough to use on entire clinic populations. The 2019 Medicare Clinical Laboratory Fee Schedule indicates that the reimbursement rate for a 9-panel immunoassay drug test is USD 65 while 1–7-panel confirmatory drug testing is USD 114 and 8–14-panel confirmatory definitive testing is USD 157 [37]. The relatively lower cost of the immunoassay test makes it more feasible and affordable than the more expensive gas chromatography mass spectrometry test in our patient population who are likely to experience significant financial constraints.

### Limitations

One limitation of this study was that the data were collected retrospectively, thereby limiting our ability to obtain detailed real-time information during the sample collection process. Future studies should utilize a prospective study design to avoid this potential limitation. Moreover, it was conducted at a single center, so the results are not easily generalizable to other centers or patient populations, particularly those with different socio-economic and demographic characteristics. The UDT was not obtained for every patient prescribed opioid medications although clinicians were encouraged to obtain the UDT within the first three visits regardless of perceived risk of NMOU. It is possible that some patients were, in effect, selected to undergo the test based on their risk profile or the clinician’s suspicion of NMOU behavior, while others were tested regardless of perceived risk. This might have increased the potential for selection bias and is a common limitation in multiple UDT studies in palliative care settings [1,12,38,39]. Lastly, the UDT in this study utilized the immunoassay technique, which has the potential for false positive results and is limited in the opioids it may detect. It was also unable to detect compounds such as oxycodone and synthetic opioids such as fentanyl and methadone. All these could have potentially resulted in false negative results. Ordering physicians often had to make further investigations to determine the aberrancy of a result. These inherent limitations of the immunoassay test could lead to an under-estimation and misrepresentation of the overall frequency of NMOU detected by abnormal UDT results in our study population.

## 5. Conclusions

Among patients receiving opioids for cancer pain at an ambulatory safety-net palliative medicine clinic who underwent immunoassay UDT, 27% and 29% of them were deemed aberrant within the first three clinic visits and during the entire study period, respectively. A significant number of them tested positive for cocaine. The findings suggest that the immunoassay UDT test might have a role in opioid therapy among patients seen in under-resourced clinical settings, especially when coupled with review of the patient record and a conversation with the patient. Future studies are needed to further examine the clinical effectiveness and benefits of immunoassay UDT in different clinical settings and to justify policy changes related to its utility in patients with cancer.

## Figures and Tables

**Figure 1 cancers-15-05663-f001:**
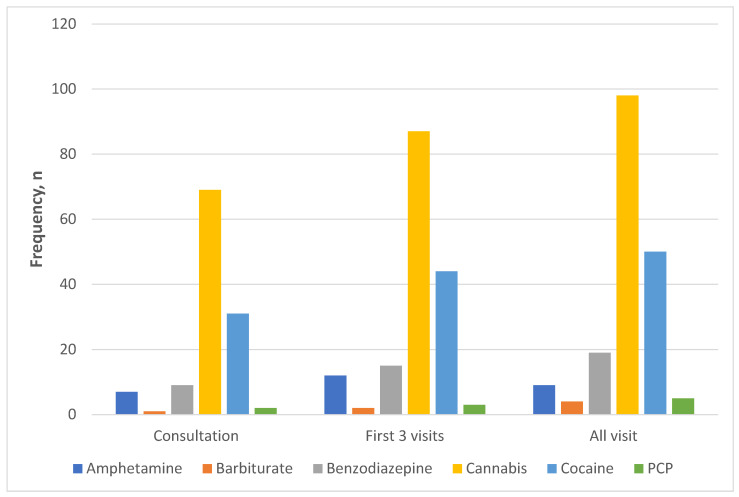
Frequency of illicit substances present among patients with aberrant urine drug test at consultation (*n* = 100), within first 3 clinic visits (*n* = 125), and for all visits (*n* = 144).

**Table 1 cancers-15-05663-t001:** Demographic and clinical characteristics of all patients seen at the palliative care clinic and those who underwent UDT within the first 3 clinic visits (*n* = 913).

Characteristic	No. of Patients (%)						
		UDT Ordered			Aberrant UDT		
	Overall (*n* = 913)	No (*n* = 458)	Yes (*n* = 455)	*p*-Value	No (*n* = 330)	Yes (*n* = 125)	*p*-Value
Age: Median (range), y	55 (18–93)	57 (18–93)	53 (24–89)	*p* < 0.001	54 (24–89)	52 (26–68)	0.02
Female sex	480 (53)	264 (58)	216 (47)	0.002	153 (46)	63 (50)	0.44
Race/ethnicity				*p* < 0.001			0.005
Hispanic, any race	425 (47)	262 (57)	163 (36)		134 (41)	29 (23)	
White, NH	206 (23)	76 (17)	130 (29)		84 (25)	46 (37)	
Black, NH	259 (28)	109 (24)	150 (33)		104 (32)	46 (37)	
Other race, NH	23 (3)	11 (2)	12 (3)		8 (2)	4 (3)	
Marital status				0.04			0.009
Single	610 (67)	291 (64)	319 (70)		220 (67)	99 (79)	
Married	303 (33)	167 (36)	13 (30)		110 (33)	26 (21)	
Cancer type				0.01			0.13
Gastrointestinal	312 (34)	177 (39)	135 (30)		94 (28)	41 (33)	
Respiratory	108 (12)	49 (11)	59 (13)		42 (13)	17 (14)	
Gynecological	106 (12)	55 (12)	51 (11)		36 (11)	15 (12)	
Genitourinary	97 (11)	44 (10)	53 (12)		36 (8)	17 (14)	
Breast	93 (10)	48 (10)	45 (10)		28 (8)	17 (14)	
Head and neck	84 (9)	28 (6)	56 (12)		46 (14)	10 (8)	
Hematological	57 (6)	26 (6)	31 (7)		27 (8)	4 (3)	
Other	56 (6)	31 (7)	25 (5)		21 (6)	4 (3)	
Cancer stage				0.20			0.57
Metastatic	614 (67)	324 (71)	290 (64)		208 (63)	82 (66)	
Locally advanced	188 (21)	80 (17)	108 (24)		76 (23)	32 (26)	
Localized	90 (10)	44 (10)	46 (10)		37 (11)	9 (7)	
Recurrent	15 (2)	7 (2)	8 (2)		6 (2)	2 (2)	
Advanced	5 (1)	2 (<1)	3 (1)		3 (1)	0	
First line	1 (<1)	1 (<1)	0		0	0	
NMOU risk factors							
History of							
Illicit drug use	161 (18)	43 (9)	118 (26)	*p* < 0.001	48 (15)	70 (56)	<0.001
Marijuana use	199 (22)	58 (13)	141 (31)	*p* < 0.001	54 (16)	87 (70)	<0.001
Tobacco use	458 (50)	191 (42)	267 (59)	*p* < 0.001	171 (52)	96 (77)	<0.001
Alcohol use	174 (19)	64 (14)	110 (24)	*p* < 0.001	74 (22)	36 (29)	0.16
Depression	159 (17)	68 (15)	91 (20)	0.04	62 (19)	29 (23)	0.30
Bipolar disorder	25 (3)	8 (2)	17 (4)	0.07	7 (2)	10 (8)	0.003
Schizophrenia	8 (1)	1 (<1)	7 (2)	0.03	3 (1)	4 (3)	0.08
Illicit drug use (in family)	17 (2)	5 (1)	12 (3)	0.09	5 (2)	7 (6)	0.02
Criminal activity	85 (9)	21 (5)	64 (14)	*p* < 0.001	24 (7)	40 (32)	<0.001
Contact with persons involved in criminal activity	59 (6)	14 (3)	45 (10)	*p* < 0.001	14 (4)	31 (25)	<0.001
Inconsistent pain presentation	15 (2)	6 (1)	9 (2)	0.43	3 (1)	6 (5)	0.008
Use for non-malignant pain	36 (4)	14 (3)	22 (5)	0.17	14 (4)	8 (6)	0.34
Others ^a^	6 (1)	1 (<1)	5 (1)	0.10	3 (1)	2 (2)	0.53
MEDD, median (IQR)	30 (10–75)	30 (10–75)	40 (15–75)	-	40 (10–75)	40 (15–70)	-

Abbreviations: UDT, urine drug test; NH, non-Hispanic; NMOU, nonmedical opioid use; MEDD, morphine equivalent daily dose (mg/day); IQR, interquartile range. ^a^ Homelessness and history of sexual abuse.

**Table 2 cancers-15-05663-t002:** Frequency and percentage of patients who were seen, completed UDT, and had aberrant UDT findings during various clinic visits.

Clinic Visit Type	Number of Patients Seen	Completed UDT, *n* (%)	Aberrant UDT, *n* (%)	Aberrant UDT Excluding Marijuana, *n* (%)
1st visit (consult)	913	356 (39)	100 (28)	85 (24)
2nd visit	639	130 (20)	47 (36)	37 (28)
3rd visit	475	75 (16)	24 (32)	21 (28)
≥4th visit	378	120 (32)	38 (32)	35 (29)
First 3 visits	913	455 (50)	125 (27)	106 (23)
All visits	913	500 (55)	144 (29)	126 (25)

**Table 3 cancers-15-05663-t003:** Multivariable regression analysis of factors associated with urine drug test ordering and aberrant UDT findings within the first three clinic visits.

Covariate	Ordering of UDT		Aberrant UDT	
	Unadjusted OR (95% CI)	Adjusted OR (95% CI)	Unadjusted OR (95% CI)	Adjusted OR (95% CI)
Age	0.98 (0.96, 0.99)	0.97 (0.96, 0.99)	0.98 (0.96, 1.00)	0.98 (0.95, 1.01)
Sex				
Male	1.51 (1.16, 1.96)	1.28 (0.89, 1.83)	0.85 (0.56, 1.28)	-
Female	1	1	1	-
Race–ethnicity				
White, non-Hispanic	2.75 (1.95, 3.88)	2.02 (1.37, 2.98)	2.53 (1.48, 4.34)	2.13 (1.03, 4.38)
Black, non-Hispanic	2.21 (1.61, 3.03)	1.86 (1.30, 2.65)	2.04 (1.20, 3.47)	1.23 (0.61, 2.45)
Other race, non-Hispanic	1.75 (0.76, 4.07)	2.20 (0.91, 5.28)	2.31 (0.65, 8.19)	3.21 (0.74, 13.87)
Hispanic, any race	1	1	1	1
Marital status				
Married	0.74 (0.56, 0.98)	0.99 (0.73, 1.35)	0.53 (0.32, 0.86)	0.84 (0.45, 1.55)
Single	1	1	1	1
Cancer diagnosis				
Respiratory	1.58 (1.01, 2.45)	1.49 (0.91, 2.44)	0.93 (0.47, 1.82)	0.57 (0.24, 1.36)
Gynecological	1.22 (0.78, 1.89)	1.35 (0.80, 2.27)	0.96 (0.47, 1.93)	0.79 (0.30, 2.05)
Genitourinary	1.58 (1.00, 2.50)	1.54 (0.92, 2.56)	1.08 (0.55, 2.14)	1.47 (0.62, 3.46)
Breast	1.22 (0.77, 1.96)	1.47 (0.85, 2.54)	1.39 (0.69, 2.82)	2.24 (0.87, 5.76)
Head and neck	2.62 (1.58, 4.35)	2.18 (1.25, 3.79)	0.49 (0.23, 1.08)	0.37 (0.14, 1.01)
Heme	1.56 (0.89, 2.76)	1.87 (0.98, 3.55)	0.34 (0.11, 1.03)	0.43 (0.10, 1.80)
Other	1.06 (0.60, 1.87)	1.03 (0.54, 1.95)	0.44 (0.14, 1.35)	0.41 (0.10, 1.69)
Gastrointestinal	1	1	1	1
Cancer stage				
Locally advanced	1.51 (1.08, 2.10)	1.58 (1.10, 2.25)	1.07 (0.66, 1.74)	-
Localized	1.17 (0.75, 1.82)	1.24 (0.77, 2.00)	0.62 (0.29, 1.34)	-
Recurrent	1.28 (0.46, 3.56)	1.63 (0.54, 4.87)	0.85 (0.17, 4.28)	-
Advanced	1.68 (0.28, 10.10)	1.52 (0.21, 11.14)	-	-
First line	-	-	-	-
Metastatic	1	1	1	-
NMOU risk factors ^a^ History of				
Illicit drug use	3.37 (2.31, 4.92)	1.81 (1.08, 3.04)	7.48 (4.69, 11.93)	3.57 (1.78, 7.13)
Marijuana use	3.09 (2.20, 4.34)	1.65 (1.09, 2.50)	11.7 (7.24, 18.91)	7.05 (3.85, 12.91)
Tobacco use	1.98 (1.52, 2.57)	1.05 (0.75, 1.47)	3.08 (1.93, 4.92)	0.97 (0.50, 1.86)
Alcohol use	1.96 (1.39, 2.75)	1.16 (0.76, 1.77)	1.40 (0.88, 2.23)	-
Depression	1.43 (1.01, 2.02)	0.96 (0.65, 1.43)	1.31 (0.79, 2.15)	-
Bipolar disorder	2.18 (0.93, 5.10)	-	4.01 (1.49, 10.79)	1.03 (0.30, 3.50)
Schizophrenia	7.12 (0.86, 58.14)	-	3.60 (0.79, 16.33)	-
Illicit drug use (in family)	2.45 (0.85, 7.01)	-	3.86 (1.20, 12.38)	1.13 (0.28, 4.55)
Criminal activity	3.40 (2.04, 5.67)	1.28 (0.61, 2.69)	6.00 (3.43, 10.51)	1.21 (0.49, 3.03)
Contact with persons involved in criminal activity	3.47 (1.88, 6.42)	1.19 (0.53, 2.69)	7.44 (3.80, 14.57)	1.48 (0.54, 4.04)
Inconsistent pain presentation	1.52 (0.54, 4.30)	-	5.50 (1.35, 22.32)	4.76 (0.70, 32.55)
Use for non-malignant pain	1.61 (0.81, 3.18)	-	1.54 (0.63, 3.77)	-
Others	5.07 (0.59, 43.54)	-	1.77 (0.29, 10.73)	-

Abbreviations: OR, odds ratio; CI, confidence interval; NMOU, nonmedical opioid use. ^a^ Reference category for each of the NMOU risk factors was no history of the individual risk factor.

## Data Availability

The data presented in the study are available on request from the corresponding author, J.M.H. The data are not publicly available because they contain information that could compromise the privacy of research subjects.

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
