# Peer review of "Immunoassay Urine Drug Testing among Patients Receiving Opioids at a Safety-Net Palliative Medicine Clinic"

_cancers, 2023, doi:10.3390/cancers15235663_

Round 1

Reviewer 1 Report (Previous Reviewer 2)

Comments and Suggestions for Authors

The authors have addressed all my comments well.

Author Response

Thank you.

Reviewer 2 Report (New Reviewer)

Comments and Suggestions for Authors

Critique

·         Much of the discussion regarding the limitations of urinary drug screening Center on false-positive tests.  The false negatives are covered rather superficially.

·         Fentanyl, methadone and buprenorphine have distinct lead different chemical structures for morphine and will not react with standard immune assays.  The to asses used in the studies would not have picked up fentanyl which is a commonly abused opioid now the major opioid in the opioid crisis.  Hence I believe that the number of abnormal urine drug tests would have been higher fentanyl had been tested.

·         Alcohol isn't be usable drug and was not acid.

·         Bupropion, ephedrine, fenfluramine, propanolol, pseudoephedrine and tyramine among other are cross reactive to amphetamines.  I personally have had an practice patients who have had positive amphetamine screen who actually were not using amphetamines but 1 of the cross reactive drugs.

·         The glucuronidated benzodiazepines may be missed in both assays.  Both assays  detect the parent compound but both assays are relatively insensitive to the metabolites.  Hence the misuse of benzodiazepines may have been missed.

·         The authors mention in the limitations that they were unable to detect fentanyl but I think particularly with the population that they deal with this is a very important issue.

·         Another weakness of the paper is that the authors did not check if the urines had been adulterated or were truly urine by measuring urine temperature, specific gravity or creatinine content.  Temperature is relatively easy to do and is inexpensive as is specific gravity.

·         The success of detecting cocaine is well known with little cross reactivity between existing drugs and the cocaine metabolite benzoylecgomine

·         A minor point as a bath salts in synthetic cannabinoids in addition to alcohol were not screened and commonly abused.

·         In light of this I would an estimate that the numbers of individuals with abnormal urine drug toxicology would be higher than what the authors have detected with their immuno assay.  I would also venture to say that it would be very important to pick up fentanyl since combinations of fentanyl another opioids particularly heroin are synergistically in both respiratory depression and reduction of cerebral blood flow.

Author Response

Thank you very much for taking the time to review our manuscript. Please find the detailed responses below and the corresponding revisions/corrections highlighted/in track changes in the re-submitted files

Comments and Suggestions for Authors

REVIEWER #1

The authors have addressed all my comments well.

RESPONSE: Thank you very much for your approval. We believe our manuscript has been greatly improved partly due to your great initial comments.

REVIEWER #2

  1. Much of the discussion regarding the limitations of urinary drug screening Center on false-positive tests.  The false negatives are covered rather superficially.

RESPONSE: Thank you. Based on your comments, we have now added further discussions of false-negative results by including the following on page 4, line 172, “On the other hand, false-negative results usually occur if a sample has a low drug concentration, or the test has a relatively high cutoff calibration based on the vendor.[17] . Most immunoassays can only recognize classes of drugs (class assays) and are unable to distinguish between drugs in the same class. They also miss compounds such as oxycodone and synthetic opioids such as fentanyl and methadone.[23] All these can lead to false-negative results.

  1. Fentanyl, methadone and buprenorphine have distinct lead different chemical structures for morphine and will not react with standard immune assays.  The to asses used in the studies would not have picked up fentanyl which is a commonly abused opioid now the major opioid in the opioid crisis.  Hence I believe that the number of abnormal urine drug tests would have been higher fentanyl had been tested.

RESPONSE: We agree with the reviewer’s comments. These are all inherent limitations of the immunoassay test. As already mentioned, we have added the statements above one page 4, line 172 of the manuscript to clarify this issue. We have also clarified this in the limitation section by making the following edits on page 10, line 341, “All these could have potentially resulted in false-negative results. Ordering physicians of-ten had to make further investigations to determine the aberrancy of a result.  These inherent limitations of the immunoassay test could lead to an under-estimation and mis-representation of the overall frequency of NMOU detected by abnormal UDT results in our study population”

  1. Alcohol isn't be usable drug and was not acid.

RESPONSE: We are so sorry, but we could not understand the point you were trying to make. We assume you were trying to say that ‘alcohol is a usable substance and was not measured.’ If so, then we’ve provided a response in #10 below, thank you

  1. Bupropion, ephedrine, fenfluramine, propanolol, pseudoephedrine and tyramine among other are cross reactive to amphetamines.  I personally have had an practice patients who have had positive amphetamine screen who actually were not using amphetamines but 1 of the cross reactive drugs.

RESPONSE: We agree with the reviewer’s comments and acknowledge that numerous substances can cross-react with opioids and other agents and therefore result in false-positive results. We are unable to list all of them in the manuscript and only provided some examples in view of the focus of this paper and being cognizant of potential journal word limitations. Based on the reviewer’s comments we have revised the following statements on page 4, line 160 to include many of the substances the reviewer listed, “The prescribing clinicians consulted the literature which described potential false positive or false negative results that may be encountered using the UDT.[19, 21, 22] For example, consumption of substances such as methylphenidate, trazodone, tyramine, labetalol, propranolol, bupropion, ephedrine, and pseudoephedrine may all lead to a positive reading for amphetamine.[23]

On page 4, line 178, we have also included the following statement and provided further references for readers who would want more information on this topic, “Details of substances that can potentially result in false-positive and false-negative results during clinical urine drug testing can be found elsewhere.[23]”

  1. The glucuronidated benzodiazepines may be missed in both assays.  Both assays  detect the parent compound but both assays are relatively insensitive to the metabolites.  Hence the misuse of benzodiazepines may have been missed.

RESPONSE: Thanks for your comments. We agree with the reviewer and acknowledge that this is yet another limitation which unfortunately is inherent to the use of immunoassays in clinical practice, as elaborated in our manuscript.

  1. The authors mention in the limitations that they were unable to detect fentanyl but I think particularly with the population that they deal with this is a very important issue.

RESPONSE: Thank you very much. We agree that inability to detect fentanyl is a real limitation in clinical practice. In general, for a situation like this, it is more preferrable to use the confirmatory UDT tests or order specific immunoassay tests which specifically detect fentanyl in urine. However, due to certain constraints, this might not be achievable in clinical settings with resource limited resources which serve the poor, uninsured, underinsured population such as ours, where the confirmatory test might not be readily affordable. This paper acknowledges these challenges and seeks to point out to readers that in the absence of the more sophisticated confirmatory tests, the immunoassay test might have some role, albeit not 100% perfect.

  1. Another weakness of the paper is that the authors did not check if the urines had been adulterated or were truly urine by measuring urine temperature, specific gravity or creatinine content.  Temperature is relatively easy to do and is inexpensive as is specific gravity.

RESPONSE: Thank you. The tests we order measures all the properties that you’ve listed above. We would like to draw your attention to the fact that this was already covered on page 4, line 181 of our manuscript, “In order to minimize the impact on our interpretation of UDT results by potential patient dilution or substitution of samples submitted, a urinalysis, or a urine creatinine, was often ordered along with the UDT. A urine pH of 3 -11, a specific gravity of 1.002- 1.020, or a creatinine of  5 mg/dl indicate an unadulterated urine sample.  Collection of the samples was not observed, so use of another person’s urine was also possible.

  1. The success of detecting cocaine is well known with little cross reactivity between existing drugs and the cocaine metabolite benzoylecgomine

RESPONSE: Thank you for your comment. You’re correct that in very rare conditions, there is a slim possibility that there could be cross-reactivity between cocaine, it’s metabolites and a substance such as salicylate but this is quite rare. Predominantly, a positive cocaine UDT test is quite conclusive.

  1. A minor point as a bath salts in synthetic cannabinoids in addition to alcohol were not screened and commonly abused.

RESPONSE: Thank you. The immunoassay test did not screen for bath salts since this is not part of the standard agents that most clinical immunoassay tests are designed to detect. We also typically do not routinely test for alcohol because traditionally, alcohol is detected by measuring urinary alcohol which has significant limitations. Because alcohol is rapidly metabolized, it is not detectable unless it has been very recently ingested. There is therefore a higher likelihood of a negative screen and patients attempting to hide their alcohol use can easily avoid alcohol for 8–12 hours before the urine screen sample is collected.

Some reference laboratories can also detect ethyl glucuronide and ethyl sulfate, two alcohol metabolites that can be detected for about 2-3 days following alcohol ingestion, but this requires a level of sophistication which is not easily affordable by resource-limited settings such as ours.

  1. In light of this I would an estimate that the numbers of individuals with abnormal urine drug toxicology would be higher than what the authors have detected with their immuno assay.  I would also venture to say that it would be very important to pick up fentanyl since combinations of fentanyl another opioids particularly heroin are synergistically in both respiratory depression and reduction of cerebral blood flow.

RESPONSE: We appreciate the reviewer’s comments. As mentioned above, we have acknowledged the under-estimation of abnormal UDS results in our manuscript on page 12, line 350 “These inherent limitations of the immunoassay test could lead to an under-estimation and misrepresentation of the overall frequency of NMOU detected by abnormal UDT results in our study population.” We also acknowledge the reviewer’s concerns about the inability of the test to detect fentanyl and have attempted to explained that in responses above in query #6. Thank you very much

This manuscript is a resubmission of an earlier submission. The following is a list of the peer review reports and author responses from that submission.

Round 1

Reviewer 1 Report

Comments and Suggestions for Authors

Thank you for submitting the manuscript. The topic is certainly of interest. However, I would like to better understand what the practical implications are, given that we find ourselves in a palliative care setting. What should toxicological investigations, and possible substance abuse, lead to? Patients' life expectancy is limited. I wonder what the benefit should be for the clinician.

Kind Regards

Author Response

Thank you for submitting the manuscript. The topic is certainly of interest. However, I would like to better understand what the practical implications are, given that we find ourselves in a palliative care setting. What should toxicological investigations, and possible substance abuse, lead to? Patients' life expectancy is limited. I wonder what the benefit should be for the clinician.

RESPONSE: Thank you for your great comments. We have further explained the implications of this in the palliative care setting by including the following statement on page 2, line 62, “This has become particularly more relevant in palliative oncology settings because with the early integration of the palliative care model into oncologic care, palliative care clinicians are caring for patients earlier in the disease trajectory and therefore encountering an increasing number of patients with chronic pain receiving opioids. Clinical evidence suggests that 1 in 5 of such patients might be at risk for NMOU.” We also added the following on page 6, line 262, “It is important to note that the use of UDS in the palliative care setting might be of more relevance among ambulatory palliative care patients with relatively longer survival than among those close to the end of life who have very short life expectancy.”

Reviewer 2 Report

Comments and Suggestions for Authors

This is an interesting study on immunoaassay urine drug testing among patients receiving opioids at a safety net palliative medicine clinic. I have several comments to improve the manuscript further:

1. In the introduction, the term NMOU is defined well, but it could be useful to provide a bit more context or examples to illustrate the implications or manifestations of NMOU in palliative care settings specifically.

2. It would be helpful to expand on the significance of the study’s objectives, highlighting the implications of understanding the role and effectiveness of immunoassay UDT in managing cancer pain in outpatient palliative care clinics, especially in resource-limited settings.

3. How was the diagnosis of substance use disorders confirmed? Was it based solely on UDT results, or were there additional diagnostic criteria?

4. More details are needed on how the aberrancy of UDT results was determined and what constituted ‘aberrant findings.’

5. With regards to the urine test, there should be additional clarification regarding the implications of false positives/negatives and how they were managed in this study. Also, It could be beneficial to explain why the specific substances were tested and how the detection of those substances is relevant in the context of opioid misuse.

6. The authors should address any potential ethical concerns related to privacy and confidentiality in the data collection and analysis process, given that the study involves potentially sensitive patient information.

7. The term “urine drug test” is used interchangeably with “UDT”. Please be consistent

8. There appears to be a slight inconsistency in the text: “Of 913 study patients seen in the clinic, 455 (50%) patients underwent a UDT within the first three visits” and later “Overall, a majority (500, 55%) of patients seen in the clinic underwent at least one UDT during the entire study period.” These statements could be more explicitly reconciled. If 50% underwent UDT in the first three visits, does this mean that an additional 5% underwent UDT in later visits? Clarifying such details would enhance understanding.

9. Issues related to generalizability should be highlighted further given that the study was conducted in a single center and is thus not easily generalizable to other centers or populations, particularly those with different socio-economic and demographic characteristics

10. The discussion highlights the limitations of the immunoassay UDT, particularly its inability to detect certain compounds and its potential for false positives. However, the implications of these limitations on the study's findings should be more thoroughly addressed. For instance, how might the inability to detect certain opioids affect the overall prevalence of aberrant UDT results in the study population?

11. The patient population is mostly comprised of racial and ethnic minorities who were uninsured or underinsured. While it mentions this demographic mix, it does not discuss in depth how this might have impacted the results, and how interventions might need to be tailored for such populations.

12. More exploration into the financial constraints of underinsured populations and the implications on the feasibility and accessibility of routine UDTs in such settings would be beneficial. The difference in costs between the various tests is mentioned, but the discussion does not go deep into how this affects the choice of test and the subsequent impact on patient care and study results.

13. The study is retrospective, and its limitations in obtaining detailed real-time information during the sample collection process are recognized, but the need for more prospective studies to corroborate these findings should be emphasized more.

14. The article briefly touches upon the exclusion of cancer-related chronic pain from certain guidelines but does not expand on the implications of such exclusions on policy and clinical practice, and whether the study's findings support a change in such policies.

Author Response

This is an interesting study on immunoaassay urine drug testing among patients receiving opioids at a safety net palliative medicine clinic. I have several comments to improve the manuscript further:

  1. In the introduction, the term NMOU is defined well, but it could be useful to provide a bit more context or examples to illustrate the implications or manifestations of NMOU in palliative care settings specifically.

RESPONSE:  Thank you very much for your suggestions. We have now provided more examples of NMOU in page 2, line 53, “It is also characterized by behaviors such as excessive, unjustifiable use of opioids or self-escalation of opioid dosage.” We have also explained further the implications of this in the palliative care setting by including the following statement on page 2, line 62, “This has become particularly more relevant in palliative oncology settings because with the early integration of the palliative care model into oncologic care, palliative care clinicians are caring for patients earlier in the disease trajectory and therefore encountering an increasing number of patients with chronic pain receiving opioids. Clinical evidence suggests that 1 in 5 of such patients might be at risk for NMOU.

  1. It would be helpful to expand on the significance of the study’s objectives, highlighting the implications of understanding the role and effectiveness of immunoassay UDT in managing cancer pain in outpatient palliative care clinics, especially in resource-limited settings.

RESPONSE:  Thank you very much for your comments. We have further expanded on the significance of the study’s objective by including the following statement on page 2, line 88, “Successful examination of UDT abnormality rates using the immunoassay test will un-derscore its significance and utility in routine opioid therapy especially in re-source-limited settings where this test might be the only available or most viable option.”

  1. How was the diagnosis of substance use disorders confirmed? Was it based solely on UDT results, or were there additional diagnostic criteria?

RESPONSE:  Thank you very much. In this study, we did not confirm substance use disorder since as you rightly indicated, that would require use of the DSM-5 diagnostic criteria. Our study however indicated that abnormal UDT might be suggestive of NMOU, which has no definitive diagnostic criteria in literature.

  1. More details are needed on how the aberrancy of UDT results was determined and what constituted ‘aberrant findings.’

RESPONSE:  Thank you for your great comment. We have now clarified what constitutes aberrant findings in our study by adding the following statement on page 4, line 152, “For this study, aberrant results were determined based on any of the following: unexpected presence of unprescribed opioids, unexpected absence of prescribed opioids, or presence of illicit drugs in the urine.”

  1. With regards to the urine test, there should be additional clarification regarding the implications of false positives/negatives and how they were managed in this study. Also, It could be beneficial to explain why the specific substances were tested and how the detection of those substances is relevant in the context of opioid misuse.

RESPONSE:  Thank you very much. Based on your insightful comment, we have now clarified on page 4, line 152 that, “False positive and negative UDT result information was ultimately discarded and not utilized in guiding further therapeutic decisions for patients in the clinic.” Also, we did clarify on page 9, line 266 that testing for illicit drugs was relevant because, “concurrent use of illicit drugs and opioids can result in increased morbidity and mortality, and that substance use of one drug such as cocaine is often accompanied by misuse of other substances such as opioids.”

  1. The authors should address any potential ethical concerns related to privacy and confidentiality in the data collection and analysis process, given that the study involves potentially sensitive patient information.

RESPONSE:  Thank you. We agree with the reviewer regarding the sensitive nature of patient information. All confidentiality and privacy procedures were followed in this study. De-identified data was used during data analysis to ensure patient privacy. The final protocol was reviewed by the institutional review board of UTHealth and Harris Health Systems who granted waiver of informed consent and authorization to conduct this study because patients were at no more than minimal risk in this study due to its retrospective nature. IRB approval was mentioned on page3, line 103. We also clarified on page 3, line 114 that, “Efforts were made during the data collection process to maintain the confidentiality and privacy of study subjects in view of the sensitive nature of the health information obtained.”

  1. The term “urine drug test” is used interchangeably with “UDT”. Please be consistent

RESPONSE:  Thank you for bringing this to our attention. We have made the edits accordingly. We used UDT as an abbreviation for urine drug test throughout the manuscript, so both are the same. There were also times when we deliberately used “urine drug testing” to indicate the testing process and this is different from the times when we used UDT to refer to the urine drug test itself.

  1. There appears to be a slight inconsistency in the text: “Of 913 study patients seen in the clinic, 455 (50%) patients underwent a UDT within the first three visits” and later “Overall, a majority (500, 55%) of patients seen in the clinic underwent at least one UDT during the entire study period.” These statements could be more explicitly reconciled. If 50% underwent UDT in the first three visits, does this mean that an additional 5% underwent UDT in later visits? Clarifying such details would enhance understanding.

Thank you very much. We have now clarified that statement on page 6, line 215, “The majority of patients seen in the clinic underwent at least one UDT (455, 50% during the first 3 visits and 500, 55% during the entire study period)”

  1. Issues related to generalizability should be highlighted further given that the study was conducted in a single center and is thus not easily generalizable to other centers or populations, particularly those with different socio-economic and demographic characteristics

Thank you very much. Based on your suggestion, we have now clarified on page 10, line 335 that the study “was conducted at a single center so the results are not easily generalizable to other centers or patient populations, particularly those with different socio-economic and demographic characteristics.”

  1. The discussion highlights the limitations of the immunoassay UDT, particularly its inability to detect certain compounds and its potential for false positives. However, the implications of these limitations on the study's findings should be more thoroughly addressed. For instance, how might the inability to detect certain opioids affect the overall prevalence of aberrant UDT results in the study population?

Thank you for your great comment. You’re correct that this inherent limitation of the immunoassay test could potentially affect the prevalence of nonmedical opioid use detected by abnormal UDT in this population. We have now expanded on this limitation by adding on page 10, line 347, “This inherent limitation of the immunoassay test could lead to misrepresentation of the overall frequency of NMOU detected by abnormal UDT in our study population.”

  1. The patient population is mostly comprised of racial and ethnic minorities who were uninsured or underinsured. While it mentions this demographic mix, it does not discuss in depth how this might have impacted the results, and how interventions might need to be tailored for such populations.

Thank you for your comment. We acknowledge that we are unable to determine from this study whether our demographic mix impacted the UDS results of the study and agree that further studies might need to explore that. We’ve expanded the discussion surrounding the demographic make up and its association with UDS ordering by including the following on page 9, page 279, “Future studies are needed to ascertain whether low social determinants of heath are key predictors of NMOU and OUD. Particularly, race or ethnicity has particularly not been found to be a risk factor for opioid misuse although data in both palliative care and non-palliative care populations have revealed disparities or inequities in UDS ordering disproportionately affecting minoritized patients”

  1. More exploration into the financial constraints of underinsured populations and the implications on the feasibility and accessibility of routine UDTs in such settings would be beneficial. The difference in costs between the various tests is mentioned, but the discussion does not go deep into how this affects the choice of test and the subsequent impact on patient care and study results.

Thank you very much for your insightful comments. We agree that the cost of the test is particularly relevant in our patient population. We did discuss it briefly on page 10 and have now added the following statement on page 10, line 334, “The relatively lower cost of the immunoassay test makes it more feasible and affordable than the more expensive gas chromatography mass spectrometry test in our patient population who are likely to experience significant financial constraints.”

  1. The study is retrospective, and its limitations in obtaining detailed real-time information during the sample collection process are recognized, but the need for more prospective studies to corroborate these findings should be emphasized more.

Thank you. We have now added on page 10, line 334, “Future studies should utilize a prospective study design in order to address this potential limitation.”

  1. The article briefly touches upon the exclusion of cancer-related chronic pain from certain guidelines but does not expand on the implications of such exclusions on policy and clinical practice, and whether the study's findings support a change in such policies.

Thank you very much for this comment. It is true that most of these guidelines exclude cancer-related pain although emerging data indicate that NMOU is similarly prevalent in the cancer patient population. While we believe that cancer patients would also benefit from such guidelines, we are unable to make such strong assertion based solely on our retrospective study and believe that more studies with more methodologically rigorous designs will be needed in order to draw robust conclusions and justify a change in those policies. We have now revised our conclusion and it now reads on page 10, line361, “Future studies are needed to further examine the clinical effectiveness and benefits of immunoassay UDT in different clinical settings and to justify policy changes related to its utility in patients with cancer.”